# Circulating Levels of SMPDL3B Define Metabolic Endophenotypes and Subclinical Kidney Alterations in Myalgic Encephalomyelitis

**DOI:** 10.3390/ijms26188882

**Published:** 2025-09-12

**Authors:** Bita Rostami-Afshari, Wesam Elremaly, Neil R. McGregor, Katherine Jin Kai Huang, Christopher W. Armstrong, Anita Franco, Christian Godbout, Mohamed Elbakry, Rim Abdelli, Alain Moreau

**Affiliations:** 1Department of Biochemistry and Molecular Medicine, Faculty of Medicine, Université de Montréal, Montreal, QC H3T 1J4, Canada; 2Viscogliosi Laboratory in Molecular Genetics of Musculoskeletal Diseases, Azrieli Research Center, CHU Sainte-Justine, Montreal, QC H3T 1C5, Canada; 3Open Medicine Foundation ME/CFS Collaborative Center, CHU Sainte-Justine/Université de Montréal, Montreal, QC H3T 1C5, Canada; 4ICanCME Research Network, Azrieli Research Center, CHU Sainte-Justine, Montreal, QC H3T 1C5, Canada; 5Department of Biochemistry and Molecular Biology, Bio21 Molecular Science and Biochemistry Institute, 30 Flemington Road, Parkville, VIC 3010, Australia; 6Faculty of Medicine, Dentistry and Health Sciences, University of Melbourne, Flemington Road, Parkville, VIC 3010, Australia; 7The Open Medicine Foundation ME/CFS Collaborative Research Centre, Bio21 Molecular Science and Biochemistry Institute, University of Melbourne, Melbourne, VIC 3010, Australia; 8Biochemistry Section, Chemistry Department, Faculty of Science, Tanta University, Tanta 31527, Gharbia Governorate, Egypt; 9Faculty of Medicine, Université Laval, Quebec, QC G1V 0A6, Canada; 10Department of Stomatology, Faculty of Dentistry, Université de Montréal, Montreal, QC H3T 1J4, Canada

**Keywords:** SMPDL3B, myalgic encephalomyelitis, renal clearance, metabolomics, 1,5-anhydroglucitol, sex differences, urinary biomarkers, glomerular barrier, lipid metabolism

## Abstract

Myalgic Encephalomyelitis (ME) is a complex, multisystem disorder with poorly understood pathophysiological mechanisms. SMPDL3B, a membrane-associated protein expressed in renal podocytes, is essential for lipid raft integrity and glomerular barrier function. We hypothesize that reduced membrane-bound SMPDL3B may contribute to podocyte dysfunction and impaired renal physiology in ME. To investigate this, we quantified soluble SMPDL3B in plasma and urine as a surrogate marker of membrane-bound SMPDL3B status and assessed renal clearance and plasma metabolomic profiles. In a cross-sectional study of 56 ME patients and 16 matched healthy controls, ME patients exhibited significantly lower urine-to-plasma ratios of soluble SMPDL3B and reduced renal clearance, suggesting podocyte-related abnormalities. Plasma metabolomics revealed dysregulation of metabolites associated with renal impairment, including succinic acid, benzoic acid, phenyllactic acid, 1,5-anhydroglucitol, histidine, and citrate. In ME patients, plasma SMPDL3B levels inversely correlated with 1,5-anhydroglucitol concentrations and renal clearance. Multivariable modeling identified the urine-to-plasma SMPDL3B ratio as an independent predictor of clearance. Female ME patients showed more pronounced SMPDL3B alterations, reduced clearance, and greater symptom severity. Non-linear associations between soluble SMPDL3B and lipid species further suggest systemic metabolic remodeling. These findings support soluble SMPDL3B as a potential non-invasive biomarker of renal-podocyte involvement in ME, highlighting sex-specific differences that may inform future therapeutic strategies.

## 1. Introduction

Myalgic Encephalomyelitis (ME), also known as Chronic Fatigue Syndrome (CFS), is a complex, debilitating illness characterized by unexplained persistent fatigue, cognitive impairment, post-exertional malaise (PEM), dysautonomia, and unrefreshing sleep [1]. In addition to these hallmark symptoms, many patients report widespread pain, gastrointestinal issues, sore throat, tender lymph nodes, and headaches [2]. Symptom severity ranges from mild to profoundly disabling, with some individuals housebound or bedridden for extended periods [2]. Despite its widespread impact and prevalence, ME remains poorly understood, with no definitive diagnostic tests or approved treatments. A critical challenge in the field is the identification of objective molecular markers that can aid in understanding the disease’s underlying mechanisms and enable stratified approaches to diagnosis and therapy.

Emerging evidence suggests that ME is a multisystemic condition characterized by metabolic disturbances, immune dysregulation, and potential organ-specific perturbations, such as mild renal involvement [3]. While overt renal dysfunction, such as end-stage kidney disease, is a well-established contributor to fatigue in clinical practice [4], our study focuses instead on subclinical alterations that elude conventional diagnostic assessments yet may contribute to the pathophysiology of ME. In this context, soluble Sphingomyelin Phosphodiesterase Acid-Like 3B (SMPDL3B) has recently gained attention as a biomarker candidate. SMPDL3B is a lipid-modulating enzyme involved in sphingomyelin metabolism [3,4], a pathway critical for maintaining cell membrane structure and regulating cellular functions such as apoptosis, inflammation, and immune signaling. A recent study has shown that elevated levels of soluble SMPDL3B are associated with greater disease severity in ME, suggesting a functional link between SMPDL3B dysregulation and the clinical manifestations of the illness [5]. Importantly, SMPDL3B also plays a recognized role in renal physiology, particularly in podocytes, which are specialized cells that preserve glomerular filtration barrier integrity. In kidney disease models, altered SMPDL3B expression has been linked with cytoskeletal disorganization, impaired lipid raft signaling, and increased glomerular permeability [6,7]. These findings suggest that SMPDL3B may serve as a mediator linking systemic metabolic imbalance and renal stress in ME. Furthermore, its broader involvement in inflammatory signaling and lipid regulation [8,9] positions it as a promising molecular proxy for investigating organ-specific perturbations in ME. Given the marked sex disparity in disease prevalence [10] and severity, with females disproportionately affected, sex-specific biological mechanisms also warrant investigation, particularly in relation to SMPDL3B and kidney-related metabolic patterns.

Therefore, this study aims to evaluate soluble SMPDL3B levels in plasma and urine as surrogate markers of renal function in ME, and to investigate their association with broader systemic metabolic indicators and potential sex-based differences. By integrating biomarker analysis with renal and metabolic profiling, we sought to determine whether alterations in SMPDL3B levels reflect subclinical kidney changes and contribute to the multisystemic pathophysiology of ME.

## 2. Results

### 2.1. Clinical and Demographic Characteristics of Participants

The clinical and demographic characteristics of study participants are summarized in Table 1. The study cohort included 56 individuals diagnosed with ME and 16 sedentary healthy controls (HC). The groups were frequency-matched by age, sex, and body mass index (BMI). The mean age of ME participants was 51.0 ± 1.4 years, comparable to the HC group (49.0 ± 3.0 years; *p* = 0.60). BMI was also similar between groups (ME: 25.0 ± 0.7 kg/m^2^; HC: 25.0 ± 1.1 kg/m^2^; *p* = 0.60). The ME group consisted of 37 females (66.1%) and 19 males (33.9%), whereas the control group had equal representation with 8 females (50.0%) and 8 males (50.0%). The average disease duration among ME patients was 10.8 ± 1.9 years, indicative of a chronically affected population.

### 2.2. Associations Between Soluble SMPDL3B, 1,5-Anhydrosorbitol, and Renal Function in Patients with ME

We first compared conventional renal function markers and soluble SMPDL3B concentrations between ME patients and healthy controls. Plasma creatinine levels did not differ significantly between ME and HC groups (ME: 0.116 ± 0.003 mM; HC: 0.110 ± 0.010 mM; *p* = 0.364), suggesting preserved gross renal function (Figure 1a). However, the urinary-to-plasma ratio of soluble SMPDL3B was significantly reduced in ME patients (2.641 ± 0.32) versus controls (4.552 ± 0.96; *p* = 0.021) (Figure 1b). Correspondingly, estimated renal clearance was decreased in ME (60.171 ± 1.85 mL/min/1.73 m^2^) compared to HC (69.466 ± 5.07 mL/min/1.73 m^2^; *p* = 0.035) (Figure 1c). Moreover, soluble SMPDL3B levels were inversely correlated with clearance (r = −0.445, *p* = 0.001; Figure 1d) but were absent in controls (Figure 1e). These consistent inverse associations suggest that elevated plasma SMPDL3B may reflect increased cleavage of its membrane-bound form, potentially via PI-PLC (phosphatidylinositol-specific phospholipase C) activity as previously reported, and may be linked to renal and metabolic dysfunction in ME. To explore the mechanistic links between soluble SMPDL3B and renal physiology, we examined the associations between plasma SMPDL3B levels and 1,5-anhydrosorbitol, a marker of tubular reabsorption and estimated kidney clearance. ME patients exhibited significantly lower plasma levels of 1,5-anhydrosorbitol compared to healthy controls (Figure 1f), consistent with impaired tubular reabsorption. Within the ME cohort, plasma 1,5-anhydrosorbitol was inversely correlated with soluble SMPDL3B (Spearman’s r = −0.585, *p* < 0.0001; Figure 1g), a relationship not observed in controls (Figure 1h).

To assess potential confounding factors, we examined the use of medications. Among the 56 ME patients, 12 were treated with medications known to influence renal function, including non-steroidal anti-inflammatory drugs (NSAIDs) [11]. However, renal clearance did not differ significantly between medicated and non-medicated patients (Appendix A), and exclusion of these individuals did not alter the statistical significance of the clearance difference between ME patients and controls. We also evaluated the potential impact of comorbidities frequently reported in ME, including allergy (39.3%), depression (42.9%), fibromyalgia (30.4%), and anxiety (32.1%) (Appendix A). None of these conditions were associated with significant differences in renal clearance within the ME cohort. Moreover, no significant correlation was observed between illness duration and renal clearance, suggesting that renal alterations may occur early in the disease course or follow a non-linear progression.

### 2.3. Urinary-to-Plasma SMPDL3B Ratio as an Independent Predictor of Renal Function

To evaluate the potential of SMPDL3B dynamics as a biomarker, multiple linear regression analysis was performed using renal clearance as the dependent variable and the urinary-to-plasma SMPDL3B ratio as the primary predictor. In the ME cohort, the model was significant (F-statistic (6, 49) = 14.47, *p* < 0.0001), with a coefficient of determination (R^2^) of 0.644 (adjusted R^2^ of 0.60). The urinary-to-plasma SMPDL3B ratio emerged as an independent predictor of renal clearance (B = 1.087, SE = 0.513, *p* = 0.039). Additional significant covariates included plasma creatinine (B = −328.6, SE = 63.62, *p* < 0.0001), 1,5-anhydrosorbitol (B = 0.0001, SE = 3.96 × 10^−5^, *p* = 0.006), and age (B = −0.586, SE = 0.125, *p* < 0.0001). Neither illness duration nor sex reached statistical significance (Table 2). Age was significantly and negatively associated with clearance (B = −0.586, SE = 0.125, *p* < 0.0001). All model assumptions were satisfied, variance inflation factors (VIFs) were below 5, and residuals followed a normal distribution, supporting the SMPDL3B ratio as a reliable marker of renal function.

### 2.4. Sex-Specific Differences in SMPDL3B Levels, Renal Function and ME Symptoms

We identified significant sex-specific differences in both molecular and clinical parameters within the ME cohort, which were not observed among healthy controls. Female ME patients exhibited a markedly lower urinary-to-plasma ratio of soluble SMPDL3B compared to male ME patients (Figure 2a). This sex-based disparity was absent in the control group, suggesting a disease-specific alteration in SMPDL3B regulation.

Similarly, renal function, as assessed by clearance measures, was significantly reduced in female ME patients relative to males (Figure 2b), while no significant differences were observed among healthy individuals. These biomarker differences were mirrored in clinical symptomatology: both female and male ME patients reported significantly more severe autonomic, endocrine, and immune-related symptoms compared to their sex-matched controls (Figure 2c). Notably, within the ME group, female patients consistently reported higher overall symptom severity than male patients. No sex-based differences in symptom scores were observed in the control population.

### 2.5. Sex-Specific Correlations Between Soluble SMPDL3B and Renal Metabolites

To further investigate the role of soluble SMPDL3B as a proxy for subtle kidney dysfunction, we performed sex-stratified correlation analyses involving both plasma and urinary SMPDL3B levels, along with selected plasma metabolites. This approach aimed to assess the systemic and renal handling of SMPDL3B and to identify sex-specific metabolic associations. Spearman’s rank correlation analyses were conducted between soluble SMPDL3B and three metabolites, citrate, hippurate, and threonine, selected for their relevance to renal physiology and their emerging roles in ME pathophysiology (Table 3). Distinct, and often contrasting, sex-specific patterns were observed.

For plasma SMPDL3B, citrate showed a non-significant negative correlation in the overall ME cohort and in females but a significant positive correlation in males (r = 0.50, *p* = 0.03), indicating a sex-dependent reversal in association. Hippurate and threonine were positively correlated with plasma SMPDL3B in the overall ME cohort (hippurate: r = 0.29, *p* = 0.03; threonine: r = 0.27, *p* = 0.05), and these associations remained significant in females (hippurate: r = 0.33, *p* = 0.04; threonine: r = 0.35, *p* = 0.03), but not in males.

For urinary SMPDL3B, citrate again showed a sex-dependent reversal: a significant positive correlation in females (r = 0.384, *p* = 0.022) and a significant negative correlation in males (r = −0.577, *p* = 0.006). Hippurate and threonine were positively correlated with urinary SMPDL3B in the overall cohort (hippurate: r = 0.308, *p* = 0.021; threonine: r = 0.305, *p* = 0.022). In females, hippurate remained significantly correlated (r = 0.375, *p* = 0.026), while threonine did not. Neither metabolite showed significant correlations in males.

### 2.6. Plasma Metabolite Alterations Associated with Renal Dysfunction in ME

To further characterize the metabolic profile associated with ME and potential subclinical kidney involvement, we analyzed plasma levels of metabolites linked to renal function and pathways commonly dysregulated in kidney disease. Several metabolites were significantly altered in ME patients compared to healthy controls (Table 4), with a predominant trend toward decreased concentrations. Specifically, ME patients exhibited significantly lower plasma levels of succinic acid (FC = 0.53, *p* = 7.68 × 10^−6^), benzoic acid (FC = 0.67, *p* = 0.001), phenyllactic acid (FC = 0.57, *p* = 0.001), 1,5-anhydrosorbitol (FC = 0.66, *p* = 0.012), L-tryptophan (FC = 0.74, *p* = 0.032), L-glutamine (FC = 0.73, *p* = 0.035), and L-kynurenine (FC = 0.72, *p* = 0.042). These associations remained significant after false discovery rate (FDR) correction. These reductions suggest potential disruptions in renal clearance and metabolic regulation.

Benzoic acid and phenyllactic acid, which are typically gut-derived and renally excreted, were reduced, indicating possible alterations in gut microbiota metabolism or renal excretion. Phenyllactic acid, known to inhibit succinate dehydrogenase, may reflect microbial metabolite accumulation during organ dysfunction [12]. The observed decrease in 1,5-anhydrosorbitol, a marker of tubular reabsorption, further supports the presence of renal impairment. Additionally, lower levels of L-tryptophan, L-glutamine, and L-kynurenine indicate dysregulated amino acid metabolism, a common feature of chronic disease states. Interestingly, succinic acid, a central intermediate in the citric acid cycle, was markedly reduced, potentially reflecting mitochondrial dysfunction, a well-documented aspect of ME pathophysiology [12]. In contrast, citrate levels were slightly elevated (FC = 1.25, *p* = 0.002), possibly indicating altered organic acid handling or citric acid cycle activity.

### 2.7. Renal and Lipidomic Signatures Across SMPDL3B-Defined Endophenotypes in ME

To identify biologically distinct ME subtypes, we stratified patients into three endophenotypes based on plasma SMPDL3B concentrations, as shown in Figure 3: low (<15 ng/mL), intermediate (16–46 ng/mL), and high (≥47 ng/mL). A clear gradient was observed in urine-to-plasma SMPDL3B ratios, with the high SMPDL3B group showing reduced urinary excretion and lower ratios (Figure 3a), suggesting impaired renal elimination. Kidney clearance was lowest in the high SMPDL3B group (Figure 3b), and plasma 1,5-anhydrosorbitol levels were also lowest (Figure 3c), indicating compromised tubular function.

To assess lipid metabolism, we examined correlations between soluble SMPDL3B and specific lipid species across endophenotypes. Lipid profiling revealed distinct patterns: PE 38:5 was lowest in the intermediate SMPDL3B group (Figure 3d), PC 40:4, and CE 20:3 were highest in the low SMPDL3B group (Figure 3e,f). In the low SMPDL3B group, all six lipids (Cer(d18:1/24:1), LPC(P-18:1), PC(P-34:2), SM 40:2, SM 42:2, and SM 44:3) showed strong positive correlations with soluble SMPDL3B (r = 0.73−0.94, all *p* < 0.05) (Table 5). In contrast, the high SMPDL3B group showed strong negative correlations (r = −0.71 to −0.88, all *p* < 0.05), while the intermediate group showed no significant associations (Table 5). These associations remained significant after false discovery rate (FDR) correction. This bidirectional shift suggests a non-linear modulatory role of soluble SMPDL3B on membrane lipid metabolism and supports its utility in defining ME subtypes with distinct renal and metabolic profiles.

## 3. Discussion

This study presents novel evidence implicating the dynamics of soluble SMPDL3B as a potential biomarker of renal and metabolic dysfunction in individuals with ME. Our findings demonstrate a consistent reduction in the urinary-to-plasma ratio of SMPDL3B in ME patients, despite normal creatinine levels, suggesting early or subclinical renal impairment. This alteration was associated with reduced renal clearance and was accompanied by specific metabolic alterations, including decreased 1,5-anhydrosorbitol and amino acid metabolites, which support the hypothesis of subtle tubular and glomerular involvement. Notably, the urinary-to-plasma SMPDL3B ratio independently predicted kidney clearance in multivariate analysis, reinforcing its potential as a sensitive and pathophysiologically relevant marker.

SMPDL3B is primarily expressed in podocytes, where its membrane-bound form maintains cytoskeletal architecture and sphingolipid balance. In models of diabetic kidney disease, excessive SMPDL3B expression impairs ceramide-1-phosphate synthesis and insulin receptor signaling, accelerating podocyte injury and glomerular decline [6]. Similarly, SMPDL3B dysregulation has been implicated in Alport syndrome and focal segmental glomerulosclerosis [6,13]. Our findings extend these observations to ME, suggesting that increased plasma SMPDL3B may indicate enhanced cleavage of its membrane-bound form, a process possibly driven by PI-PLC activity, as previously described in ME [5]. Reduced urinary SMPDL3B may therefore reflect impaired excretion or depletion of SMPDL3B from renal membranes, potentially affecting podocyte function and glomerular barrier integrity. This interpretation is further supported by inverse correlations between plasma SMPDL3B and both renal clearance and 1,5-anhydrosorbitol, a marker of tubular reabsorption [14].

Sex-specific analyses revealed that female ME patients exhibited more pronounced reductions in renal clearance and urinary-to-plasma SMPDL3B ratios compared to males, suggesting that sex may modulate SMPDL3B expression or renal processing in ME. This was accompanied by significantly decreased kidney clearance in females, who also reported greater severity of autonomic, endocrine, and immune-related symptoms. Notably, these sex differences were absent in healthy controls, highlighting a disease-specific biological divergence. Such disparities may reflect fundamental differences in SMPDL3B expression and shedding, or renal excretory capacity, between sexes. Furthermore, correlation analyses revealed sex-dependent reversals in associations between SMPDL3B and key metabolites, most notably citrate, which correlated positively with urinary SMPDL3B in females but negatively in males. Citrate is a known regulator of mitochondrial function and acid-base homeostasis and is reabsorbed in the proximal tubule [15]. This reversal may indicate divergent compensatory mechanisms or mitochondrial responses to renal stress in males and females.

Our metabolomic profiling identified several significantly altered metabolites in ME patients, converging on mitochondrial function, amino acid metabolism, and renal excretion, and offering deeper insights into ME pathophysiology. ME patients exhibited lower plasma concentrations of multiple key metabolites, including succinic acid, benzoic acid, phenyllactic acid, 1,5-anhydrosorbitol, L-tryptophan, L-glutamine, and L-kynurenine. These alterations are consistent with emerging features of ME: impaired renal excretion, dysbiosis-related changes in microbial metabolites, and widespread mitochondrial dysfunction [16,17,18]. The marked reduction in succinic acid, a core intermediate of the tricarboxylic acid (TCA) cycle, strongly supports previous hypotheses implicating mitochondrial energy deficits in ME [19]. This suggests a fundamental impairment in cellular energy metabolism. This energy deficit provides a plausible explanation for the paradox of reduced kidney clearance with normal creatinine levels, suggesting that creatinine production may be lowered in ME. Additionally, reduced levels of tryptophan and kynurenine, often observed in chronic inflammation and immune dysregulation, further support the presence of systemic metabolic disturbances. Interestingly, elevated plasma citrate levels in ME patients may reflect altered organic acid handling or compensatory responses in citric acid cycle flux, warranting further investigation. Other metabolite changes highlight the complex interplay between the gut and kidneys. For example, hippurate, a gut microbiota-derived uremic toxin typically cleared by the kidneys, often accumulates in chronic kidney disease (CKD) and serves as an indicator of reduced renal excretory capacity [20]. Given the proposed alterations in gut microbiota composition and increased intestinal permeability in ME, which contribute to systemic inflammation and metabolic dysregulation [17,21], hippurate’s dysregulation in ME patients reinforces the notion of compromised gut-kidney axis function. Similarly, phenyllactic acid, which can accumulate during organ dysfunction and inhibit succinate dehydrogenase, is particularly relevant to gut-kidney-mitochondrial crosstalk [22].

The observed reductions in essential amino acids, such as L-tryptophan, L-glutamine, and L-kynurenine, further underscore the profound dysregulation of amino acid metabolism. These amino acids are crucial for immune modulation, mitochondrial activity, and renal homeostasis [23]. Specifically, disruption of the kynurenine pathway, evidenced by altered L-tryptophan and L-kynurenine levels, is believed to play a significant role in neurodegenerative and cognitive disorders, directly linking it to symptomology in ME [24]. Additionally, glutamine depletion could impair nitrogen balance and energy metabolism [25], contributing to the broader metabolic challenges seen in ME. Threonine, an essential amino acid vital for protein synthesis and immune regulation, particularly in the gut via mucin synthesis [26], also showed altered plasma levels. This may reflect disturbed amino acid metabolism or renal handling [23], both of which are pertinent to the immune and metabolic dysfunction associated with ME. Together, these widespread metabolic perturbations support the hypothesis that ME is characterized by a multisystemic biochemical phenotype involving energy imbalance, immune dysregulation, and subtle yet significant renal impairment. Understanding these interconnected metabolic pathways is crucial for developing targeted interventions for ME.

Stratification by plasma SMPDL3B levels uncovered three biologically distinct endophenotypes with divergent renal and lipidomic profiles. This approach revealed a complex and impactful relationship between circulating SMPDL3B and both renal and metabolic dysfunction. We found that patients in the high SMPDL3B group exhibited the lowest urinary-to-plasma ratio of SMPDL3B levels, kidney clearance, and plasma intensity of 1,5-anhydrosorbitol, indicating more pronounced renal impairment and tubular dysfunction. Interestingly, PE plasma intensity was higher in both the extremely low and high SMPDL3B phenotypes. PE species are crucial for maintaining mitochondrial membrane structure and function, and their dysregulation is linked to oxidative stress and impaired energy metabolism [27,28], both of which are key features of ME [29,30] and kidney disease [31]. Moreover, our detailed lipidomic profiling across these SMPDL3B-defined endophenotypes uncovered bidirectional disruptions in sphingolipid homeostasis. In the low SMPDL3B group, several membrane-associated lipids, including ceramide C24:1, sphingomyelin SM 42:2, and phosphatidylethanolamine PE 38:5, showed strong positive correlations with SMPDL3B. These correlations disappeared in the intermediate group, and SMPDL3B was instead negatively associated in the high SMPDL3B group. This non-linear pattern suggests that both low and high plasma SMPDL3B concentrations may reflect pathological states involving loss of membrane-bound SMPDL3B and consequent lipid disorganization. These findings are consistent with previous in vitro and in vivo models. In RAW264.7 macrophages, SMPDL3B knockdown led to decreased ceramide C24:1, supporting its role in sphingomyelin-to-ceramide conversion [8]. Similarly, in a mouse model of unilateral ureteral obstruction, progressive renal injury was associated with loss of ceramide C24:1 and multiple sphingomyelins, such as SM 42:2 [32]. Our findings extend these models to human ME patients, suggesting that enhancement of membrane-bound SMPDL3B cleavage by PI-PLC or reduced synthesis of SMPDL3B due to epigenetic mechanisms disrupts membrane lipid domains essential for podocyte signaling and mitochondrial integrity. This may explain the dual association of SMPDL3B with both lipid depletion and reduced kidney function in ME.

A major strength of this study lies in its integrative approach, combining biomarker profiling, renal function assessment, targeted metabolomics, and lipidomics to elucidate the mechanistic pathways underlying ME pathophysiology. The inclusion of sex-stratified and endophenotype-based analyses enhances clinical relevance and may facilitate the development of personalized diagnostic strategies in the future. Furthermore, the identification of soluble SMPDL3B as a candidate biomarker opens new avenues for investigating podocyte dysfunction and its systemic consequences in ME. Despite promising findings, several limitations should be acknowledged. First, the cross-sectional design precludes causal inference and limits our ability to determine the temporal onset of renal alterations. Second, although comprehensive, its size, particularly for subgroup analyses, may limit statistical power. Larger, longitudinal studies are needed to validate these observations and assess the prognostic value of SMPDL3B dynamics. Third, while SMPDL3B levels were measured in plasma and urine, direct tissue-based validation (e.g., via renal biopsy or advanced imaging) was not feasible, and future studies incorporating such methods may provide deeper mechanistic insights into the role of SMPDL3B in ME. Fourth, while NSAID use was not a statistically significant confounder in our sample (*p* = 0.112), the observed trend suggests that this factor warrants further investigation in larger, more powered studies.

In conclusion, this study provides the first evidence that altered soluble SMPDL3B distribution is associated with subclinical renal dysfunction and metabolic abnormalities in ME. The urinary-to-plasma SMPDL3B ratio emerges as a promising independent marker of kidney clearance and may serve as a useful clinical tool for detecting early renal stress in this population. Sex-specific differences and distinct SMPDL3B-defined endophenotypes underscore the heterogeneity of ME, supporting the development of precision medicine approaches. These findings underscore the interconnection between podocyte biology, systemic metabolism, and immune-neuroendocrine symptoms in ME, paving the way for future research into targeted diagnostics and interventions.

## 4. Materials and Methods

### 4.1. Sex as a Biological Variable

Although ME has a significant prevalence among females, both female and male participants were studied, which led to a skewed sample size (with more female patients). The ME group and sedentary healthy controls were frequency-matched by age, sex, and body mass index (BMI).

### 4.2. Study Populations

This cross-sectional study included a total of 72 participants: 56 individuals diagnosed with ME (37 females and 19 males) and 16 sedentary healthy controls (HC) (8 females and 8 males). ME diagnoses were confirmed using the Canadian Consensus Criteria. All participants were recruited prior to the onset of the COVID-19 pandemic, were aged 18 years or older, and self-identified as Caucasian of European ancestry. Participants in the control group were recruited based on the following inclusion criteria: (1) a self-reported sedentary lifestyle, defined as engaging in less than two hours of moderate-intensity physical activity per week; (2) no personal or family history of ME, fibromyalgia (FM), multiple sclerosis (MS), or other chronic illnesses; and (3) no current or recent (within the past month) viral or bacterial infection at the time of enrollment. To minimize potential confounding factors, controls were frequency-matched to the ME cohort based on group-level sex and age distributions. The study protocol was approved by the Institutional Review Board of CHU Sainte-Justine (protocol 4047), and all participants provided written informed consent prior to enrollment.

### 4.3. Clinical and Demographic Data Collection

Demographic information collected for all participants, including age, sex, illness duration (in years), and body mass index (BMI, calculated as kg/m^2^), is reported in Table 1.

### 4.4. Assessment of Health Status, Symptoms, and Disease Severity

Health status and symptom severity were assessed using validated, self-administered clinical questionnaires tailored to the study cohort. All participants completed three standardized instruments: the 36-Item Short Form Health Survey (SF-36), the Multidimensional Fatigue Inventory (MFI-20), and the DePaul Symptom Questionnaire (DSQ). The SF-36 is a widely used tool for evaluating general health-related quality of life, encompassing both physical and mental health domains [33]. The MFI-20 assesses five distinct dimensions of fatigue: general fatigue, physical fatigue, mental fatigue, reduced activity, and reduced motivation, with higher scores reflecting greater symptom burden [34]. The DSQ is specifically designed to evaluate ME symptomatology, categorizing responses into four major domains: cognitive dysfunction, neuroendocrine/autonomic/immune disturbances, post-exertional malaise, and sleep disruption [35]. Together, these instruments provided a multidimensional assessment of symptom severity and functional impairment, supporting stratification of clinical profiles across participants.

### 4.5. Sample Collection and Processing

Blood and urine samples were collected from all participants using standardized protocols to ensure consistency and sample integrity. Venous blood was drawn into EDTA tubes via standard venipuncture and immediately centrifuged at 216× *g* for 10 min at room temperature to separate plasma from cellular components. Urine samples were collected as clean-catch midstream specimens to minimize contamination. All plasma and urine aliquots were stored at −80 °C until further analysis to preserve analyte stability and prevent degradation.

### 4.6. Measurement of Plasma and Urinary SMPDL3B Levels

Plasma and urinary concentrations of SMPDL3B were measured using a commercially available sandwich enzyme-linked immunosorbent assay (ELISA) kit (Cat. ABX585223; Abbexa Ltd., Cambridge, UK), following the manufacturer’s protocol. All samples were analyzed in duplicate, and the mean value of the two measurements was used for statistical analysis to enhance reliability and reduce technical variability. Urinary creatinine concentrations were determined using a colorimetric assay (R&D Systems, Minneapolis, MN, USA) to control for variations in urine concentration. Urinary SMPDL3B levels were normalized to urinary creatinine and expressed as ng/mL per mg of creatinine.

### 4.7. Plasma and Urinary Metabolite Profiling by NMR Spectroscopy

Metabolomic profiling of plasma and urine samples was conducted using proton nuclear magnetic resonance [^1^H] (NMR) spectroscopy. A total volume of 250 µL of plasma and urine was used per participant. Plasma samples were prepared for NMR analysis via liquid–liquid extraction, following protocols previously described in the literature [30]. Spectral acquisition was performed using a high-resolution 700 MHz Bruker Avance HDIII spectrometer (Bruker, Billerica, MA, USA), equipped with a cryogenically cooled probe, which provided a 3- to 4-fold increase in sensitivity. A SampleJet cooled autosampler (Bruker) was used to maintain sample integrity throughout the acquisition process. Data collection adhered to standardized NMR protocols, as described in prior methodological references [30,36].

### 4.8. Plasma Metabolite Profiling by Mass Spectrometry (MS)

In addition to NMR, plasma metabolites underwent targeted profiling using Mass spectrometry (MS). Plasma samples were prepared according to the polar metabolite extraction protocols provided by Metabolomics Australia (Bio21 Molecular Science and Biochemistry Institute, Parkville, Australia). Briefly, 20 μL of plasma was added to 180 μL of ice-cold extraction solvent with internal standards (acetonitrile/methanol/water 40:40:20 with 3 μM ^13^C6-Sorbitol, 3 μM ^13^C6-Leucine, and 3 μM ^13^C5,15N1-Valine). Samples were sonicated and centrifuged (14,100× *g*, 10 min at 0 °C) to pellet precipitated protein and cell debris, and the supernatant was transferred for analysis. Additionally, 10 μL of supernatant from each sample was transferred to prepare the pooled biological quality controls (PBQCs).

Metabolite quantification was performed by Metabolomics Australia using a liquid chromatography-tandem mass spectrometry (LC-MS/MS) platform. This approach allowed for the detection and quantification of a broad range of plasma metabolites and lipids, providing complementary data to the NMR analysis. One μL of each sample, including PBQCs between every five samples, was injected on an Agilent 6545B series quadrupole time-of-flight mass spectrometer (QTOF-MS, Agilent Technologies, Santa Clara, CA, USA) using a hydrophilic column (ZIC-pHILIC). Data extraction and metabolite annotation were performed using the MassHunter Quantitative Analysis Software (Version B.09.00, Agilent Technologies), based on retention time and molecular mass.

### 4.9. Estimation of Renal Clearance

Renal clearance was estimated using the Cockcroft–Gault equation, which incorporates age, body weight, sex, and plasma creatinine concentration to approximate creatinine clearance. Sex-specific constants were applied: 1.23 for males and 1.04 for females. To account for differences in body size, the resulting clearance values were adjusted for each participant’s body surface area (BSA) and expressed in mL/min/1.73 m^2^, following standard clinical normalization practices [37].

### 4.10. Statistical Analysis

All results are presented as mean ± standard error of the mean (SEM). Group comparisons between two groups were performed using the non-parametric Mann–Whitney U test. For comparisons involving more than two groups, the Kruskal–Wallis test was used; when significant, Dunn’s post hoc test with correction for multiple comparisons was applied. For data meeting assumptions of normality and homogeneity of variance, one-way analysis of variance (ANOVA) was used, followed by Tukey’s honestly significant difference (HSD) test for post hoc pairwise comparisons. In cases where the data were non-normally distributed, the Kruskal–Wallis test and Dunn’s post hoc test were applied instead. Non-normality of continuous variables was formally confirmed using Shapiro–Wilk, Anderson–Darling, and Kolmogorov–Smirnov tests and is interpreted as a reflection of biological heterogeneity within the ME cohort rather than age-related drift.

Associations between variables were assessed using Spearman’s rank correlation coefficient due to the non-normal distribution of several parameters. A *p*-value of ≤0.05 was considered statistically significant. To explore predictors of renal clearance (mL/min/1.73 m^2^), a multiple linear regression model was constructed. The primary independent variable was the ratio of urinary to plasma soluble SMPDL3B. Additional covariates included plasma creatinine concentration, 1,5-anhydrosorbitol levels, age, illness duration, and biological sex (numerically coded as 1 = female, 2 = male). Model assumptions were rigorously evaluated: linearity was assessed using residual plots; multicollinearity was checked via Variance Inflation Factor (VIF), with VIF < 5 deemed acceptable; and the normality of residuals was evaluated using the Anderson–Darling, D’Agostino–Pearson, Shapiro–Wilk, and Kolmogorov–Smirnov tests. Homoscedasticity was visually assessed through residual versus fitted value plots.

All statistical analyses were performed using GraphPad Prism, version 8.4.3 for Mac OS (GraphPad Software, San Diego, CA, USA). Statistical power and minimum sample size requirements were determined using G*Power software (version 3.1.9.7, 17 March 2020) [38], ensuring sufficient power to detect meaningful group differences.

## Figures and Tables

**Figure 1 ijms-26-08882-f001:**
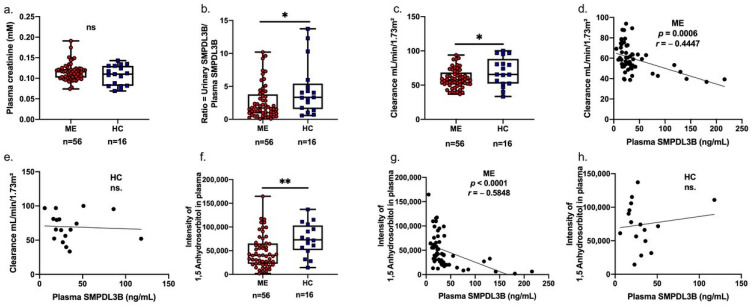
Associations between plasma-soluble SMPDL3B, 1,5-anhydrosorbitol, and kidney clearance in ME patients and healthy controls. (**a**) Box plot comparing plasma creatinine levels between Myalgic Encephalomyelitis (ME) patients (red circles) and healthy controls (HC) (blue squares), showing individual distribution, minimum, and maximum values. (**b**) Box plot comparing urinary-to-plasma ratio of soluble SMPDL3B levels between ME patients (red circles) and HC (blue squares), showing individual distribution, minimum, and maximum values. (**c**) Box plot comparing kidney clearance (mL/min/1.73 m^2^) between ME patients (red circles) and healthy controls (blue squares), showing individual distribution, minimum, and maximum values. (**d**) Scatter plot illustrating the negative correlation between plasma soluble SMPDL3B (ng/mL) and kidney clearance (mL/min/1.73 m^2^) in ME patients (Spearman’s r = −0.4447, *p* = 0.0006). (**e**) Scatter plot showing the non-significant correlation between plasma soluble SMPDL3B and kidney clearance in healthy controls. (**f**) Box plot comparing plasma 1,5-anhydrosorbitol levels between ME patients (red circles) and HC (blue squares), showing individual distribution, minimum, and maximum values. (**g**) Scatter plot illustrating the negative correlation between plasma 1,5-anhydrosorbitol and plasma soluble SMPDL3B (ng/mL) in ME patients (Spearman’s r = −0.5848, *p* = 0.0001). (**h**) Scatter plot showing the non-significant correlation between plasma 1,5-anhydrosorbitol and plasma soluble SMPDL3B in healthy controls. Statistical comparisons between ME patients and healthy controls were performed using independent samples *t*-tests. Levels of significance are indicated as * *p* < 0.05, ** *p* < 0.01, and ns is non significant.

**Figure 2 ijms-26-08882-f002:**
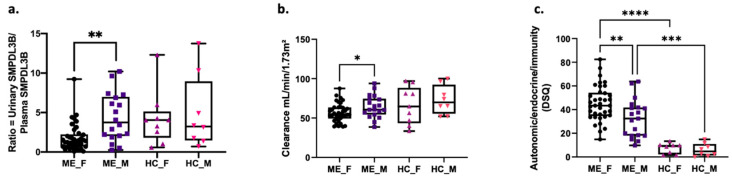
Sex-specific differences in soluble SMPDL3B, kidney clearance, and symptom severity in ME patients and healthy controls. (**a**) Ratio of urinary-to-plasma SMPDL3B (ng/mL) based on biological sex. (**b**) Kidney clearance (mL/min/1.73 m^2^) as a function of biological sex. (**c**) Severity of autonomic/endocrine/immunity scores from the DePaul symptom questionnaire (DSQ) in function of biological sex. Results are presented as box plots showing individual variations, minimum, and maximum values. Females ME patients (black circles, *n* = 37), ME male patients (dark purple squares, *n* = 19). Female healthy controls (pink-purple triangles, *n* = 8) and male healthy controls (orange downward triangles, *n* = 8). Statistical analysis was performed using ANOVA followed by Tukey’s post hoc test. Levels of significance are indicated as * *p* < 0.05, ** *p* < 0.01, *** *p* < 0.001 and **** *p* < 0.0001.

**Figure 3 ijms-26-08882-f003:**
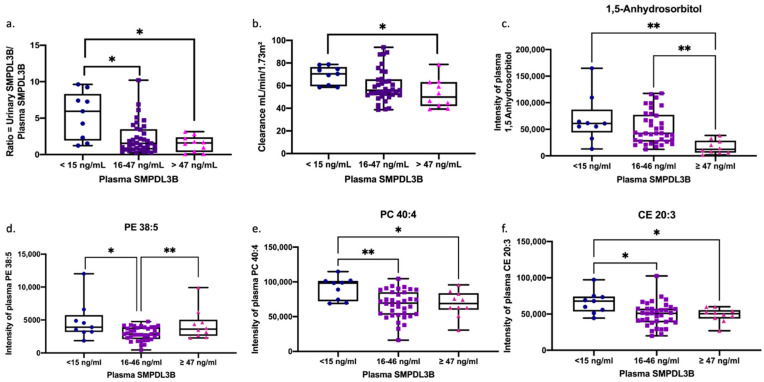
Distinct metabolic profiles across soluble SMPDL3B subgroups in ME patients. Patients were stratified into three groups based on their plasma soluble SMPDL3B levels: extremely low (<15 ng/mL), middle level (16–46 ng/mL), and extremely high (≥47 ng/mL). For each group, the ratio of urine-to-plasma SMPDL3B (**a**), the clearance (mL/min/1.73 m^2^) (**b**), the plasma levels of 1,5-anhydrosorbitol (**c**), and the intensities of plasma PE 38:5, PC 40:4 and CE 20:3 ((**d**), (**e**), and (**f**), respectively) were measured. Results are presented as box plots showing minimum, maximum, and individual variations. Statistical analysis was performed using ANOVA followed by Tukey’s post hoc test. The level of significance are indicated as * *p* < 0.05 and ** *p* < 0.01.

**Table 1 ijms-26-08882-t001:** Clinical and demographic characteristics of the participants.

Characteristics	ME (*n* = 56)	HC (*n* = 16)	*p*-Value
Sex, *n* (%)			
Female	37 (66.1%)	8 (50.0%)	
Male	19 (33.9%)	8 (50.0%)	
Age, years (mean ± SEM)	51 ± 1.4	49 ± 3.0	0.6
BMI, kg/m^2^ (mean ± SEM)	25.0 ± 0.7	25.0 ± 1.1	0.6
Illness duration, years (Mean ± SEM)	10.87 ± 1.9	N/A	N/A

Values are presented as the number of participants (*n*) for each sex, and as the mean ± standard error of the mean (SEM) for age, BMI, and illness duration. *p*-values are from independent samples *t*-tests comparing the Myalgic Encephalomyelitis (ME) and healthy control (HC) groups. N/A indicates not applicable.

**Table 2 ijms-26-08882-t002:** Multiple linear regression analysis predicting renal clearance using urinary/plasma SMPDL3B ratio in ME patients.

Variable	Unstandardized Coefficient (B)	Standard Error (SE)	Standardized Coefficient (β) *	t-Value	*p*-Value	95% Confidence Interval for B
Model: Clearance						
Overall Model Fit: R^2^ = 0.644	Adjusted R^2^ = 0.600	F-statistic (6, 48) = 14.47	*p* < 0.0001			
Predictors:						
Intercept	110	10.05	N/A	10.94	<0.0001 ****	89.75 to 130.2
Creatinine Concentrations (mM)	−328.6	63.62	N/A	−5.166	<0.0001 ****	−456.5 to −200.7
Ratio SMPDL3B (urinary/plasma)	1.087	0.513	N/A	2.118	0.0394 *	0.055 to 2.118
1,5-Anhydrosorbitol	0.0001	3.96 × 10^−5^	N/A	2.86	0.0063 **	3.364 × 10^−5^ to 0.0001
Age	−0.586	0.125	N/A	−4.693	<0.0001 ****	−0.837 to −0.335
Illness duration	0.161	0.097	N/A	1.665	0.1025	−0.033 to 0.355
Sex	4.456	2.717	N/A	1.64	0.1075	−1.006 to 9.919

Clearance (mL/min/1.73 m^2^) in Myalgic Encephalomyelitis (ME) patients. The overall model was highly statistically significant (F-statistic (6, 48) = 14.47, *p* < 0.0001), with an R^2^ of 0.644 and an adjusted R^2^ of 0.6002. Data are presented as unstandardized coefficients (B) and their standard errors (SE). Standardized coefficient (β) were not available in the provided output. Levels of significance are indicated as * *p* < 0.05, ** *p* < 0.01, **** *p* < 0.0001. N/A is not applicable.

**Table 3 ijms-26-08882-t003:** Spearman’s rank correlations (r) of select plasma metabolites with plasma and urinary soluble SMPDL3B levels in ME patients.

Variable (Soluble SMPDL3B vs.)	Plasma SMPDL3B All ME Patients (*n* = 56)	Plasma SMPDL3B Female ME Patients (*n* = 37)	Plasma SMPDL3B Male ME Patients (*n* = 19)	Urinary SMPDL3B All ME Patients (*n* = 56)	Urinary SMPDL3B Female ME Patients (*n* = 37)	Urinary SMPDL3B Male ME Patients (*n* = 19)
Citrate	r = −0.03 *p* = 0.84	r = −0.24 *p* = 0.15	r = 0.50 *p* = 0.03 *	r = −0.248 *p* = 0.06	r = 0.384 *p* = 0.022 *	r = −0.577 *p* = 0.006 **
Hippurate	r = 0.29 *p* = 0.03 *	r = 0.33 *p* = 0.04 *	r = 0.24 *p* = 0.32	r = 0.308 *p* = 0.021 *	r = 0.3751 *p* = 0.026 *	r = 0.391 *p* = 0.07
Threonine	r = 0.27 *p* = 0.05 *	r = 0.35 *p* = 0.03 *	r = 0.22 *p* = 0.37	r = 0.305 *p* = 0.022 *	r = 0.211 *p* = 0.22	r = 0.395 *p* = 0.07

Spearman’s rank correlations (r) of select plasma metabolites with plasma and urinary soluble SMPDL3B levels in ME patients. * *p* < 0.05, ** *p* < 0.01. *p*-values in bold indicate statistical significance.

**Table 4 ijms-26-08882-t004:** Plasma metabolite alterations related to renal function in ME patients compared to healthy controls.

Metabolite Name (HMDB ID)	ME (*n* = 56)	HC (*n* = 16)	FC	Status (ME Relative to HC)	*t*-Test	Adjusted *p*-Value
Succinic acid_HMDB0000254	80,673 ± 5369	151,759 ± 20377	0.53	Decreased	7.68 × 10^−6^	0.002
Benzoic acid_HMDB0001870	3,098,959 ± 170,374	4,658,580 ± 548,375	0.67	Decreased	0.001	0.006
Phenyllactic acid_HMDB0000779	9450 ± 630	16,711 ± 3124	0.57	Decreased	0.001	0.005
1,5-Anhydrosorbitol_HMDB0002712	48,838 ± 4615	74,131 ± 8514	0.66	Decreased	0.012	0.035
L-Tryptophan_HMDB0000929	521,597 ± 38,773	703,767 ± 76,957	0.74	Decreased	0.032	0.046
L-Glutamine_HMDB0000641	653,315 ± 52,695	893,783 ± 99,977	0.73	Decreased	0.035	0.048
L-Kynurenine_HMDB0000684	2664 ± 212	3705 ± 586	0.72	Decreased	0.042	0.047
Citrate	0.005 ± 0.0001	0.004 ± 0.0002	1.25	Increased	0.0015	0.045

Data are presented as mean ± standard error of the mean (SEM). Fold change (FC) is calculated as the mean metabolite level in ME patients divided by the mean metabolite level in healthy controls (ME_AVG/HC_AVG). The status of metabolites indicates whether the metabolite level is increased or decreased in ME patients relative to controls. *p*-values are derived from independent samples *t*-tests comparing ME patients to healthy controls. Values considered statistically significant if *p* < 0.05.

**Table 5 ijms-26-08882-t005:** Common metabolic signature in ME patients with very low and very high plasma SMPDL3B levels.

ID of Metabolites	Plasma SMPDL3B Levels <15 ng/mL			Plasma SMPDL3B Levels 16–46 ng/mL			Plasma SMPDL3B Levels ≥47 ng/mL		
	r	*p*-Value	FDR	r	*p*-Value	FDR	r	*p*-Value	FDR
Cer(d18:1/24:1)	0.796	0.026	0.043	0.003	0.985	2.463	−0.709	0.027	0.029
LPC(P-18:1)	0.726	0.035	0.045	0.089	0.608	1.013	−0.770	0.013	0.025
PC(P-34:2)	0.726	0.035	0.043	0.079	0.647	0.719	−0.781	0.011	0.024
SM 40:2	0.752	0.026	0.045	−0.029	0.867	1.495	−0.721	0.023	0.026
SM 42:2	0.761	0.023	0.049	0.218	0.201	2.010	−0.879	0.002	0.017
SM 44:3	0.936	0.001	0.022	0.235	0.167	1.392	−0.709	0.027	0.027

Spearman’s rank correlations (r) of specific plasma lipids with soluble SMPDL3B levels (within the defined ranges) in ME patients stratified by soluble SMPDL3B concentrations. Correlations (r) and *p*-values are presented for patients with very low (<15 ng/mL), middle-level (16–46 ng/mL), and very high (≥47 ng/mL) plasma soluble SMPDL3B. Values considered statistically significant if *p* < 0.05.

## Data Availability

The original contributions presented in this study are included in the article/Appendix A. Further inquiries can be directed to the corresponding authors.

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
