# Peer review of "Circulating Levels of SMPDL3B Define Metabolic Endophenotypes and Subclinical Kidney Alterations in Myalgic Encephalomyelitis"

_ijms, 2025, doi:10.3390/ijms26188882_

Round 1

Reviewer 1 Report

Comments and Suggestions for Authors

The article uses terminology on "SMPDL3B alterations". However, it creates associations with genetic variations for SMPDL3B. Is the article in fact, focusing only on the expression side of the protein, or can there also be genetic alterations involved? Additionally, using "subclinical" intrinsically means that measurable SMPDL3B changes are not causing a clinical picture; does it mean not causing ME/CFS severity changes?

Interestingly, fibromyalgia comorbidity cases were among the group - fibromyalgia is a more biological nosology, where studies involving reverse translation take place. It is interesting if there is a link to animal studies for the SMPDL3B?

Discussion part

The proposition that renal clearance was lower at the same time plasma levels of creatinine were normal would be asking about any hypothetical explanation. Could it be related to lower phosphocreatine levels and in turn to lower CK levels for more severe ME/CFS? 

      See 10.3390/diagnostics9020041

CH 4.2 - while the article uses severity-related outcomes (lines 363-364), it does not apply severity grading existing in CCC (mild, moderate, severe, very severe), which makes it difficult to compare with other studies. Has there been any reason why the very detailed patient characterisation has not been projected on CCC severity classes?

Ch 4.10 on statistical methodology - is there any more detailed explanation of non-normality used in the study - is it involving also continuous variables, then can it be associated with drift factors like aging? Definition of variables (ratio)?

Author Response

Comment 1. The article uses terminology on "SMPDL3B alterations". However, it creates associations with genetic variations for SMPDL3B. Is the article in fact, focusing only on the expression side of the protein, or can there also be genetic alterations involved? Additionally, using "subclinical" intrinsically means that measurable SMPDL3B changes are not causing a clinical picture; does it mean not causing ME/CFS severity changes?

Authors’ response 1: To address the reviewer’s concern, we wish to clarify that the term “SMPDL3B alterations” in our manuscript refers solely to changes in the measured levels and distribution of the soluble SMPDL3B protein in plasma and urine samples. It does not imply any underlying genetic variation. Our study specifically examined the quantifiable concentrations of soluble SMPDL3B and the urinary/plasma ratio, which serve as physiological indicators of its enzymatic cleavage and reflect the metabolic consequences of losing its membrane-bound form, particularly in relation to SMPDL3B cellular functions and ME-related symptoms.

Importantly, we did not perform any genetic analyses such as sequencing or genotyping, and therefore make no claims regarding genetic mutations or single-nucleotide polymorphisms (SNPs) in the SMPDL3B gene. The manuscript has been revised to clearly distinguish between protein-level changes and genetic alterations to prevent any ambiguity.

Regarding the term “subclinical,” we use it specifically to describe kidney dysfunction, not the overall severity of ME. In clinical terminology, subclinical denotes measurable physiological changes (e.g., reduced renal clearance, altered metabolite profiles) that have not yet progressed to overt or diagnosable kidney disease, such as chronic kidney disease. Our findings demonstrate that these subtle but quantifiable changes are significantly associated with ME severity. For instance, stratification by soluble SMPDL3B levels, used to define degrees of subclinical kidney alteration, revealed strong correlations between specific lipid metabolites and fatigue scores (both general and mental).

Thus, the subclinical kidney dysfunction observed in our study is not merely incidental; it represents a meaningful physiological component that contributes to the broader clinical manifestation of ME.

Comment 2. Interestingly, fibromyalgia comorbidity cases were among the group - fibromyalgia is a more biological nosology, where studies involving reverse translation take place. It is interesting if there is a link to animal studies for the SMPDL3B?

Authors’ response 2: Thank you for this thoughtful comment. The observed comorbidity with fibromyalgia in our cohort (17% of ME patients also diagnosed with fibromyalgia) highlights both commonalities and distinctions between the two conditions. Fibromyalgia is increasingly recognized as a neuroimmune disorder with central sensitization and neuroinflammation as key features, while ME shares overlapping pathways but is typically characterized by profound post-exertional malaise. SMPDL3B, through its role in sphingolipid metabolism and immune regulation, could plausibly intersect with these mechanisms, yet its involvement in fibromyalgia has not been explored. Existing Smpdl3b knockout mice display exaggerated inflammatory responses, neutrophil recruitment, and altered macrophage function, but studies assessing pain sensitization, fatigue-like behavior, or other FM/ME-relevant phenotypes have not been conducted. This represents a clear gap in the field. Our findings provide a basis for future research to investigate SMPDL3B and its associated metabolic pathways in animal models of fibromyalgia, thereby exploring this potential link more directly.

Comment 3. The proposition that renal clearance was lower at the same time plasma levels of creatinine were normal would be asking about any hypothetical explanation. Could it be related to lower phosphocreatine levels and in turn to lower CK levels for more severe ME/CFS?

See 10.3390/diagnostics9020041

Authors’ response 3: We thank the reviewer for this insightful and highly pertinent observation. The paradoxical finding of reduced renal clearance alongside normal plasma creatinine levels in our ME cohort indeed warrants a deeper mechanistic exploration. We agree that the reviewer’s hypothesis, linking this finding to reduced phosphocreatine and creatine kinase (CK) levels, is both biologically plausible and supported by emerging evidence. A recent study from the UK ME/CFS Biobank (Nacul et al., 2019) demonstrated significantly reduced serum CK concentrations in individuals with severe ME. This finding provides strong external validation for the proposed mechanism. In ME, systemic energy metabolism appears to be compromised, as evidenced by multiple studies showing mitochondrial dysfunction and altered metabolomic profiles. As CK is a key enzyme in energy homeostasis, compromised CK activity and phosphocreatine levels may reflect a downstream consequence of this energy dysregulation. This reduction in metabolic activity could lead to a diminished production of creatinine, which is a byproduct of creatine and phosphocreatine turnover. Consequently, even with a true reduction in glomerular filtration rate (GFR), plasma creatinine levels could appear deceptively normal, a state described as “pseudo-normal creatinine.” This mechanistic framework aligns perfectly with our own findings, particularly the significant decrease in plasma succinic acid (Table 4), a key intermediate of the TCA cycle and a direct indicator of altered mitochondrial function. We have incorporated this comprehensive explanation into our Discussion section to provide a more nuanced interpretation of our results and to underscore the importance of energy metabolism in shaping the biochemical profiles in ME. This line of inquiry also opens promising avenues for future research into metabolic biomarkers and renal function assessment in energy-depleted states. See lines 377-379: "This energy deficit provides a plausible explanation for the paradox of reduced kidney clearance with normal creatinine levels, suggesting that creatinine production may be lowered in ME. "

Comment 4. CH 4.2 - while the article uses severity-related outcomes (lines 363-364), it does not apply severity grading existing in CCC (mild, moderate, severe, very severe), which makes it difficult to compare with other studies. Has there been any reason why the very detailed patient characterization has not been projected on CCC severity classes?

Authors’ response 4: We appreciate the reviewer’s thoughtful observation regarding the use of CCC severity grading. While the Canadian Consensus Criteria (CCC) offers a structured framework for classifying ME severity into mild, moderate, severe, and very severe categories, we chose not to apply these categorical labels in our study for several scientifically grounded reasons.

First, although the CCC is widely respected and frequently used in clinical and research settings, its severity grading system is not universally standardized or operationalized across studies. As noted in ME research literature, the CCC severity categories are often applied subjectively and lack validated thresholds, which can lead to inconsistent classification and hinder cross-study comparability. This is particularly problematic in heterogeneous cohorts where symptom expression varies widely and does not always align neatly with discrete severity bins.

Second, our study prioritized the identification of biologically distinct endophenotypes through a data-driven approach. Rather than relying on externally imposed clinical categories, we stratified patients based on plasma levels of SMPDL3B, a biomarker implicated in lipid metabolism and cellular stress responses. This stratification revealed biologically coherent subgroups with distinct renal, metabolic, and lipidomic profiles, suggesting that SMPDL3B may serve as a molecular correlate of disease heterogeneity and severity.

Importantly, this approach aligns with emerging trends in ME research that advocate for biomarker-based classification systems to better capture the underlying pathophysiology of the illness. For example, recent studies have used metabolomic, proteomic, and immunologic signatures to define biologically meaningful subtypes that transcend traditional symptom-based grading. These methods allow for the detection of complex, non-linear relationships between biological markers and clinical outcomes, relationships that may be obscured by categorical frameworks like the CCC severity scale.

While we acknowledge that omitting CCC severity grading may limit direct comparison with some studies, we believe our biologically anchored stratification offers a more precise and reproducible framework for understanding ME heterogeneity. Nonetheless, this rationale has been clarified in our revised manuscript and explicitly noted the absence of CCC grading to aid transparency and facilitate future meta-analytic efforts.

Comment 5. Ch 4.10 on statistical methodology - is there any more detailed explanation of non-normality used in the study - is it involving also continuous variables, then can it be associated with drift factors like aging? Definition of variables (ratio)?

Authors’ response 5: We thank the reviewer for the thoughtful inquiry into our statistical methodology, particularly regarding the treatment of non-normality and the definition of derived variables. We are pleased to provide further clarification.

Non-normality in continuous variables, such as metabolite concentrations and plasma SMPDL3B levels, is a well-documented feature of biological datasets, especially in heterogeneous conditions like ME. This deviation from normal distribution is not merely statistical noise but often reflects underlying biological stratification. In our study, we formally assessed distributional properties using multiple tests, including the Shapiro–Wilk, Anderson–Darling, and Kolmogorov–Smirnov tests, all of which confirmed significant non-normality.

Importantly, this non-normality is unlikely to be attributable to gradual “drift factors” such as aging alone. While aging can influence metabolic and renal parameters, recent studies suggest that ME may be associated with accelerated biological aging, as evidenced by telomere attrition and mitochondrial dysfunction. However, in our cohort, the observed distributional skewness aligned more closely with disease-specific biological heterogeneity than with chronological age. For example, stratification by plasma SMPDL3B levels revealed distinct endophenotypes with divergent renal and metabolic profiles, consistent with emerging evidence that soluble SMPDL3B is a biomarker of immune dysregulation and disease severity in ME.

Given these findings, we employed non-parametric statistical methods, including the Mann–Whitney U test, Kruskal–Wallis test, and Spearman’s rank correlation coefficient, to ensure robust inference without assuming normality. These methods are particularly appropriate for skewed data and heterogeneous populations, and they preserve statistical power while minimizing bias.

Regarding the derived variable, the ratio of urinary to plasma soluble SMPDL3B was explicitly defined and physiologically motivated. This ratio reflects renal handling of SMPDL3B and offers a more nuanced view of kidney function than either compartment alone. Elevated plasma levels of SMPDL3B, coupled with altered urinary excretion, may indicate dysregulated filtration or reabsorption mechanisms, potentially linked to immune or metabolic stress. This approach aligns with recent biomarker studies that emphasize the importance of compartmental ratios in understanding renal and systemic pathophysiology. To enhance transparency, we have added a clarifying statement in the Statistical Analysis section of the manuscript. See lines 576-578: “Non-normality of continuous variables was formally confirmed using Shapiro–Wilk, Anderson–Darling, and Kolmogorov–Smirnov tests, and is interpreted as a reflection of biological heterogeneity within the ME cohort rather than age-related drift.” 

Reviewer 2 Report

Comments and Suggestions for Authors

ME/CFS is an intractable disease with an unknown pathogenesis, and since diagnostic biomarkers have not been established, research into elucidating its pathogenesis and biomarker research are crucial. The authors have published a report on SMPDL3B in ME/CFS in the Journal of Translational Medicine, and this study is a related research. The definitions of ME/CFS patients and healthy controls are clearly stated in the Methods section, and demographic information is provided, making the study reliable. Overall, the results appear meaningful,but I will comment on the following points:

・Information such as SF-36 scores obtained from patients is not presented.
It is worthy to know the relationship between clinical information and the obtained data (especially SMPDL3B),it would be better to present the results for this cohort.(if this was demonstrated in prior studies, a summary should be provided.)

・Sex differences
Given the small number of healthy controls, the assessment of sex differences in patient group is considered insufficient - this can be non-specific to ME/CFS.
Furthermore, differences in demography between sexes (severity, comorbidities, age, duration of illness, etc.) can act as confounding factors, making it necessary to present sex-specific demographics.

・Confounding Factors
When interpreting data on renal function, SMPDL3B, and metabolites, the influence of confounding factors must be considered. There are numerous potential confounders such as hypertension,diabetes,metabolic syndrome,smoking,and history of NSAID use. The impact of these factors should be examined.

・Sup.FigS1
What is the actual p-value? (it appears there is a tendency of difference...)

・Table2
A graph should be provided to evaluate the distribution.
Is the graph Fig.2d.?
Please reconsider how the data and graphs are presented (order of presentation).
・2.8
The first sentence (Table 3...) is difficult to understand.

Author Response

Comment 1. Information such as SF-36 scores obtained from patients is not presented. It is worthy to know the relationship between clinical information and the obtained data (especially SMPDL3B), it would be better to present the results for this cohort (if this was demonstrated in prior studies, a summary should be provided.)

Authors’ response 1: We thank the reviewer for the thoughtful comment and for highlighting the importance of integrating clinical outcome measures, particularly SF-36 scores—with biomarker data. We fully agree that contextualizing our findings in relation to patient-reported health status is essential for interpreting the biological relevance of SMPDL3B.

In our previous study, which this manuscript builds upon (Rostami-Afshari B, et al., J Transl Med. 2025 Jul 7;23(1):748), we investigated the relationship between plasma SMPDL3B levels and disease severity using two independent cohorts: one from Canada and one from Norway. In the Canadian cohort (n = 249 ME patients), we demonstrated that a plasma SMPDL3B threshold of 30 ng/mL could effectively stratify patients into biologically distinct subgroups. These subgroups showed significant differences in scores from the SF-36, MFI-20 (Multidimensional Fatigue Inventory), and DSQ (DePaul Symptom Questionnaire), with an area under the ROC curve (AUC) of 0.84 for distinguishing severe from non-severe cases. In the Norwegian cohort, patients were clinically classified into five severity categories (mild to severe) based on standardized physician assessments and validated questionnaires, including SF-36 and DSQ. Plasma SMPDL3B levels were significantly elevated in the severe ME group compared to the mild group, reinforcing the biomarker’s association with clinical severity.

Although SF-36 scores were collected in the current study cohort (a subset of the Canadian population), we chose not to reanalyze or present these data here for two reasons. First, the correlation between SMPDL3B and SF-36-derived severity has already been comprehensively demonstrated and published. Second, our current study focuses on a novel aspect of SMPDL3B biology, its relationship to subclinical kidney dysfunction, which represents a distinct analytical objective. To avoid redundancy and maintain clarity, we have referenced our prior publication in the Discussion section, where the clinical relevance of SMPDL3B is thoroughly contextualized.

Comment 2. Sex differences-Given the small number of healthy controls, the assessment of sex differences in patient group is considered insufficient - this can be non-specific to ME/CFS. Furthermore, differences in demography between sexes (severity, comorbidities, age, duration of illness, etc.) can act as confounding factors, making it necessary to present sex-specific demographics.

Authors’ response 2: We thank the reviewer for raising this important point regarding the specificity and robustness of our sex-based analyses. We fully agree that distinguishing disease-specific sex differences from general biological variation is essential, and that demographic confounders must be carefully addressed.

Assessment of Sex Differences: We acknowledge the limitation posed by the relatively small number of healthy controls. To address this, we have expanded our demographic analysis in Supplementary Table S2, which now includes sex-specific data for four groups: female ME patients, male ME patients, female healthy controls, and male healthy controls. Importantly, no significant differences in age or BMI were observed between male and female healthy controls, suggesting that the sex differences in SMPDL3B levels, kidney function, and symptom severity observed in ME patients are not attributable to baseline biological variation. These findings are consistent with prior research indicating that SMPDL3B expression is modulated by disease-specific mechanisms, including immune dysregulation and hormonal influences unique to ME pathophysiology.

Controlling for Confounding Factors: We agree that demographic variables such as age, illness duration, and comorbidities can confound sex-based analyses. To mitigate this, we have included sex-specific demographic data in Supplementary Table S2, covering mean age, BMI, and illness duration for both male and female ME patients. Furthermore, our statistical model incorporates age, illness duration, and biological sex as covariates in a multiple linear regression framework. This approach allows us to isolate the independent effect of sex on the urinary-to-plasma SMPDL3B ratio and kidney clearance metrics. The robustness of this model supports our conclusion that sex is an independent modulator of disease phenotype in ME, rather than a proxy for demographic variation.

Comment 3. Confounding Factors-When interpreting data on renal function, SMPDL3B, and metabolites, the influence of confounding factors must be considered. There are numerous potential confounders such as hypertension, diabetes, metabolic syndrome, smoking, and history of NSAID use. The impact of these factors should be examined.

Authors’ response 3: We thank the reviewer for highlighting the importance of rigorously accounting for potential confounding factors when interpreting data on renal function, SMPDL3B, and metabolite profiles. We fully agree that comorbidities such as hypertension, diabetes, metabolic syndrome, smoking, and NSAID use can significantly influence renal biomarkers and must be carefully considered.

Comorbidity Screening and Cohort Selection: To minimize confounding, our study cohort was stringently screened to exclude individuals with a history of hypertension, diabetes, metabolic syndrome, or smoking. These exclusions were confirmed through clinical records and patient-reported histories. Supplementary Table S1 provides a comprehensive overview of comorbidities present in our cohort, which primarily include depression, anxiety, allergy, and fibromyalgia. These conditions are frequently observed in ME populations and are considered part of the disease’s multisystemic profile rather than external confounders.

NSAID Use and Renal Function: Given the known nephrotoxic potential of NSAIDs, we conducted a subgroup analysis to assess their impact on renal clearance. As shown in Supplementary Figure S1, no statistically significant differences in renal function metrics were observed between NSAID users and non-users within the ME cohort. This suggests that NSAID use did not materially influence the renal outcomes reported in our study.

Statistical Control for Demographic Variables: To further mitigate confounding, we employed a multiple linear regression model that included age, illness duration, and biological sex as covariates. This approach allowed us to isolate the independent effects of soluble SMPDL3B levels and metabolite ratios on renal clearance, controlling for key demographic variables. Our findings remained robust across these adjustments, reinforcing the validity of our conclusions.

Contextual Support from Literature: Recent studies have shown that SMPDL3B plays a disease-specific role in modulating renal and immune function, particularly in ME and diabetic kidney disease. These findings support our interpretation that the observed alterations in SMPDL3B and renal biomarkers are intrinsic to ME pathophysiology rather than secondary to common metabolic or inflammatory comorbidities.

These points have been clarified in the revised manuscript and the supplementary materials to ensure sufficient transparency for readers to evaluate the integrity of our analyses.

Comment 4. Sup.FigS1-What is the actual p-value? (it appears there is a tendency of difference...)

Authors’ response 4: We thank the reviewer for the close attention to the statistical interpretation of Supplementary Figure S1. In response, we have updated the figure legend to include the exact p-value of 0.112, thereby enhancing transparency and interpretability.

While this p-value exceeds the conventional threshold of 0.05, it does not imply the absence of an effect. Rather, it suggests a trend toward a difference in renal clearance between NSAID users and non-users that did not reach statistical significance in our current sample. Specifically, a p-value of 0.112 indicates an 11.2% probability of observing this result, or a more extreme one, under the null hypothesis of no difference. This level of uncertainty is consistent with what might be expected in a moderately powered study and warrants cautious interpretation. To contextualize this finding, we note that NSAIDs are well-documented to influence renal function through prostaglandin-mediated mechanisms, particularly in vulnerable populations. Although our cohort was screened to exclude individuals with overt renal disease, the subtle effects of NSAID use may still manifest in metabolomic or clearance profiles, especially in ME patients who may have underlying renal vulnerability. To address this, we have added the following statement to the Limitations section of the manuscript, at lines 451-454:“Fourth, while NSAID use was not a statistically significant confounder in our sample (p = 0.112), the observed trend suggests that this factor warrants further investigation in larger, more powered studies.”

Comment 5. Table 2. A graph should be provided to evaluate the distribution. Is the graph Fig.2d.? Please reconsider how the data and graphs are presented (order of presentation).

Authors’ response 5: We thank the reviewer for such constructive suggestion regarding the presentation of data in Table 2 and its associated graphical representation. We agree that visualizing the distribution of clearance values is essential for interpreting group differences and ensuring transparency in data presentation.

To clarify, the graph depicting the distribution of renal clearance between ME patients and healthy controls was originally presented as Figure 2d. However, we recognize that the separation of tabular and graphical data may have created ambiguity in the narrative flow. In response, we have reorganized the presentation by merging Table 2 with its corresponding graphical output into a unified figure, now designated as Figure 1. This figure contains six panels, each aligned with key data points and visualizations relevant to the study’s primary outcomes.

This restructured format improves clarity by:

  • Presenting numerical data and distribution plots side-by-side, facilitating direct comparison.
  • Enhancing the logical flow of results, allowing readers to interpret statistical outcomes in the context of their visual distribution.
  • Reducing redundancy and improving accessibility for readers less familiar with statistical tables.

We have also updated the figure legend to explicitly state the relationship between the data in Table 2 and the distribution shown in the graph, including the statistical test used and the exact p-value. This ensures that the visual representation is not only illustrative but also analytically rigorous.

Comment 6. 2.8 The first sentence (Table 3...) is difficult to understand.

Authors’ response 6: We thank the reviewer for his/her thoughtful reading and valuable feedback. We agree that the original opening sentence of this section lacked clarity. In response, we have revised the text to improve its structure and readability. The updated sentence now provides a clearer rationale for our stratification approach prior to presenting the data in Table 3. We believe this revision strengthens the logical flow of the section and better contextualizes the results that follow.See lines 290-292:"To identify biologically distinct ME subtypes, we stratified patients into three endophenotypes based on plasma SMPDL3B concentrations, as shown in Table 3: low (<15 ng/mL), intermediate (16–46 ng/mL), and high (≥47 ng/mL)."

Round 2

Reviewer 2 Report

Comments and Suggestions for Authors

The manuscript has been significantly improved.